# Exact inference in structured prediction

**Kevin Bello**
Department of Computer Science
Purdue Univeristy
West Lafayette, IN 47906, USA
kbellome@purdue.edu

**Jean Honorio**
Department of Computer Science
Purdue Univeristy
West Lafayette, IN 47906, USA
jhonorio@purdue.edu

## Abstract

Structured prediction can be thought of as a simultaneous prediction of multiple labels. This is often done by maximizing a score function on the space of labels, which decomposes as a sum of pairwise and unary potentials. The above is naturally modeled with a graph, where edges and vertices are related to pairwise and unary potentials, respectively. We consider the generative process proposed by Globerson et al. (2015) and apply it to general connected graphs. We analyze the structural conditions of the graph that allow for the exact recovery of the labels. Our results show that exact recovery is possible and achievable in polynomial time for a large class of graphs. In particular, we show that graphs that are bad expanders can be exactly recovered by adding small edge perturbations coming from the Erdős-Rényi model. Finally, as a byproduct of our analysis, we provide an extension of Cheeger's inequality.

## 1 Introduction

Throughout the years, structured prediction has been continuously used in multiple domains such as computer vision, natural language processing, and computational biology. Examples of structured prediction problems include dependency parsing, image segmentation, part-of-speech tagging, named entity recognition, and protein folding. In this setting, the input $X$ is some observation, e.g., social network, an image, a sentence. The output is a labeling $y$, e.g., an assignment of each individual of a social network to a cluster, or an assignment of each pixel in the image to foreground or background, or the parse tree for the sentence. A common approach to structured prediction is to exploit local features to infer the global structure. For instance, one could include a feature that encourages two individuals of a social network to be assigned to different clusters whenever there is a strong disagreement in opinions about a particular subject. Then, one can define a posterior distribution over the set of possible labelings conditioned on the input. Some classical methods for learning the parameters of the model are conditional random fields (Lafferty et al. 2001) and structured support vector machines (Taskar et al. 2003, Tsochantaridis et al. 2005, Altun & Hofmann 2003). In this work we will focus in the inference problem and assume that the model parameters have been already learned.

In the context of Markov random fields (MRFs), for an undirected graph $\mathcal{G} = (\mathcal{V}, \mathcal{E})$, one is interested in finding a solution to the following inference problem:

$$\max_{\boldsymbol{y} \in \mathcal{M}^{|\mathcal{V}|}} \sum_{v \in \mathcal{V}, m \in \mathcal{M}} c_v(m) \mathbb{1}[y_v = m] + \sum_{(u,v) \in \mathcal{E}, m_1, m_2 \in \mathcal{M}} c_{u,v}(m, n) \mathbb{1}[y_u = m, y_v = n], \quad (1)$$

where $\mathcal{M}$ is the set of possible labels, $c_u(m)$ is the cost of assigning label $m$ to node $v$, and $c_{u,v}(m, n)$ is the cost of assigning $m$ and $n$ to the neighbors $u, v$ respectively.[1] Similar inference problems arise

in the context of statistical physics, sociology, community detection, average case analysis, and graph partitioning. Very few cases of the general MRF inference problem are known to be exactly solvable in polynomial time. For example, Chandrasekaran et al. (2008) showed that (1) can be solved exactly in polynomial time for a graph $\mathcal{G}$ with low treewidth via the junction tree algorithm. While in the case of Ising models, Schraudolph & Kamenetsky (2009) showed that the inference problem can also be solved exactly in polynomial time for planar graphs via perfect matchings. Finally, polynomial-time solvability can also stem from properties of the pairwise potential, under this view, the inference problem can be solved exactly in polynomial time via graph cuts for binary labels and sub-modular pairwise potentials (Boykov & Veksler 2006).

Despite the intractability of maximum likelihood estimation, maximum a-posteriori estimation, and marginal inference for most models in the worst case, the inference task seems to be easier in practice than the theoretical worst case. Approximate inference algorithms can be extremely effective, often obtaining state-of-the-art results for these structured prediction tasks. Some important theoretical and empirical work on approximate inference include (Foster et al. 2018, Globerson et al. 2015, Kulesza & Pereira 2007, Sontag et al. 2012, Koo et al. 2010, Daumé et al. 2009).

In particular, Globerson et al. (2015) analyzes the hardness of approximate inference in the case where performance is measured through the Hamming error, and provide conditions for the minimum-achievable Hamming error by studying a generative model. Similar to the objective (1), the authors in (Globerson et al. 2015) consider unary and pairwise noisy observations. As a concrete example (Foster et al. 2018), consider the problem of trying to recover opinions of individuals in social networks. Suppose that every individual in a social network can hold one of two opinions labeled by $-1$ or $+1$. One observes a measurement of whether neighbors in the network have an agreement in opinion, but the value of each measurement is flipped with probability $p$ (pairwise observations). Additionally, one receives estimates of the opinion of each individual, perhaps using a classification model on their profile, but these estimates are corrupted with probability $q$ (unary observations). Foster et al. (2018) generalizes the work of Globerson et al. (2015), who provides results for grid lattices, by providing results for trees and general graphs that allow tree decompositions (e.g., hypergrids and ring lattices).

Note that the above problem is challenging since there is a *statistical* and *computational* trade-off, as in several machine learning problems. The statistical part focuses on giving highly accurate labels while ignoring computational constraints. In practice this is unrealistic, one cannot afford to wait long times for each prediction, which motivated several studies on this trade-off (e.g., Chandrasekaran & Jordan (2013), Bello & Honorio (2018)).

However, while the statistical and computational trade-off appears in general, an interesting question is whether there are conditions for when recovery of the true labels is achievable in polynomial time. That is, conditions for when the Hamming error of the prediction is zero and can be obtained efficiently. The present work addresses this question. In contrast to (Globerson et al. 2015, Foster et al. 2018), we study the sufficient conditions for exact recovery in polynomial time, and provide high probability results for *general* families of undirected connected graphs, which we consider to be a novel result to the best of our knowledge. In particular, we show that weak-expander graphs (e.g., grids) can be exactly recovered by adding small perturbations (edges coming from the Erdős-Rényi model with small probability). Also, as a byproduct of our analysis, we provide an extension of Cheeger's inequality (Cheeger 1969). Finally, another work in this line was done by Chen et al. (2016), where the authors consider exact recovery for edges on sparse graphs such as grids and rings. However, (Chen et al. 2016) consider the case where one has multiple i.i.d. observations of edge labels. In contrast, we focus on the case where there is a single (noisy) observation of each edge and node in the graph.

## 2 Notation and Problem Formulation

This section introduces the notation used throughout the paper and formally defines the problem under analysis.

Vectors and matrices are denoted by lowercase and uppercase bold faced letters respectively (e.g., $\boldsymbol{a}, \boldsymbol{A}$), while scalars are in normal font weight (e.g., $a$). Moreover, random variables are written in upright shape (e.g., $\mathbf{a}, \mathbf{A}$). For a random vector $\mathbf{a}$, and a random matrix $\mathbf{A}$, their entries are denoted by $\mathbf{a}_i$ and $\mathbf{A}_{i,j}$ respectively. Indexing starts at $1$, with $\mathbf{A}_{i,:}$ and $\mathbf{A}_{:,i}$ indicating the $i$-th row and $i$-th

column of $\mathbf{A}$ respectively. Finally, sets and tuples are both expressed in uppercase calligraphic fonts and shall be distinguished by the context. For example, $\mathcal{R}$ will denote the set of real numbers.

We now present the inference task. We consider a similar problem setting to the one in (Globerson et al. 2015), with the only difference that we consider general undirected graphs. That is, the goal is to predict a vector of $n$ node labels $\boldsymbol{y} = (y_1, \ldots, y_n)^\top$, where $y_i \in \{+1, -1\}$, from a set of observations $\mathbf{X}$ and $\mathbf{c}$, where $\mathbf{X}$ and $\mathbf{c}$ correspond to corrupted measurements of edges and nodes respectively. These observations are assumed to be generated from a ground truth labeling $\boldsymbol{y}^*$ by a generative process defined via an undirected connected graph $\mathcal{G} = (\mathcal{V}, \mathcal{E})$, an edge noise $p \in (0, 0.5)$, and a node noise $q \in (0, 0.5)$. For each edge $(u, v) \in \mathcal{E}$, the edge observation $\mathbf{X}_{u,v}$ is independently sampled to be $y_u^* y_v^*$ (*good edge*) with probability $1 - p$, and $-y_u^* y_v^*$ (*bad edge*) with probability $p$. While for each edge $(u, v) \notin \mathcal{E}$, the observation $\mathbf{X}_{u,v}$ is always 0. Similarly, for each node $u \in \mathcal{V}$, the node observation $\mathbf{c}_u$ is independently sampled to be $y_u^*$ (*good node*) with probability $1 - q$, and $-y_u^*$ (*bad node*) with probability $q$. Thus, we have a *known* undirected connected graph $\mathcal{G}$, an *unknown* ground truth label vector $\boldsymbol{y}^* \in \{+1, -1\}^n$, and noisy observations $\mathbf{X} \in \{-1, 0, +1\}^{n \times n}$ and $\mathbf{c} \in \{-1, +1\}^n$, and our goal is to predict a vector label $\boldsymbol{y} \in \{-1, +1\}^n$.

**Definition 1** (Biased Rademacher variable). *Let* $\mathbf{z}_p \in \{+1, -1\}$ *such that* $P(\mathbf{z}_p = +1) = 1 - p$, *and* $P(\mathbf{z}_p = -1) = p$. *We call* $\mathbf{z}_p$ *a biased Rademacher random variable with parameter* $p$ *and expected value* $1 - 2p$.

From the definition above, we can write the edge observations as $\mathbf{X}_{u,v} = y_u^* y_v^* \mathbf{z}_p^{(u,v)} \mathbb{1}\big[(u, v) \in \mathcal{E}\big]$, where $\mathbf{z}_p^{(u,v)}$ is a biased Rademacher with parameter $p$. While the node observation is $c_u = y_u^* \mathbf{z}_q^{(u)}$, where $\mathbf{z}_q^{(u)}$ is a biased Rademacher with parameter $q$.

Given the generative process, we aim to solve the following optimization problem, which is based on the maximum likelihood estimator that returns the label $\arg\max_{\boldsymbol{y}} P(\mathbf{X}, \boldsymbol{y})$ (see Globerson et al. (2015)):

$$\max_{\boldsymbol{y}} \quad \frac{1}{2} \boldsymbol{y}^\top \mathbf{X} \boldsymbol{y} + \alpha \mathbf{c}^\top \boldsymbol{y} \quad \text{subject to} \quad y_i = \pm 1, \tag{2}$$

where $\alpha = \log \frac{1-q}{q} / \log \frac{1-p}{p}$. In general, the above combinatorial problem is NP-hard to compute (e.g., see for results on grids (Barahona 1982)). Our goal is to find what structural properties of the graph $\mathcal{G}$ suffice to achieve, with high probability, exact recovery in polynomial time.

# 3 On Exact Recovery of Labels

Our approach consists of two stages, similar in spirit to (Globerson et al. 2015). We first use only the quadratic term from (2), which will give us two possible solutions, and then as a second stage, the linear term is used to decide the best between these two solutions.

## 3.1 First Stage

We analyze a semidefinite program (SDP) relaxation to the following combinatorial problem (3), motivated by the techniques in (Abbe et al. 2016).

$$\max_{\boldsymbol{y}} \quad \frac{1}{2} \boldsymbol{y}^\top \mathbf{X} \boldsymbol{y} \quad \text{subject to} \quad y_i = \pm 1, \tag{3}$$

We denote the degree of node $i$ as $\Delta_i$, and the maximum node degree as $\Delta_{\max} = \max_{i \in \mathcal{V}} \Delta_i$. For any subset $\mathcal{S} \subset \mathcal{V}$, we denote its complement by $\mathcal{S}^C$ such that $\mathcal{S} \cup \mathcal{S}^C = \mathcal{V}$ and $\mathcal{S} \cap \mathcal{S}^C = \emptyset$. Furthermore, let $\mathcal{E}(\mathcal{S}, \mathcal{S}^C) = \{(i, j) \in \mathcal{E} \mid i \in \mathcal{S}, j \in \mathcal{S}^C \text{ or } j \in \mathcal{S}, i \in \mathcal{S}^C\}$, i.e., $|\mathcal{E}(\mathcal{S}, \mathcal{S}^C)|$ denotes the number of edges between $\mathcal{S}$ and $\mathcal{S}^C$.

**Definition 2** (Edge Expansion). *For a set* $\mathcal{S} \subset \mathcal{V}$ *with* $|\mathcal{S}| \leq {}^n\!/_2$, *its edge expansion,* $\phi_{\mathcal{S}}$, *is defined as:* $\phi_{\mathcal{S}} = |\mathcal{E}(\mathcal{S}, \mathcal{S}^C)| / |\mathcal{S}|$. *Then, the edge expansion of a graph* $\mathcal{G} = (\mathcal{V}, \mathcal{E})$ *is defined as:* $\phi_{\mathcal{G}} = \min_{\mathcal{S} \subset \mathcal{V}, |\mathcal{S}| \leq {}^n\!/_2} \phi_{\mathcal{S}}$.

In the literature, $\phi_{\mathcal{G}}$ is also known as the *Cheeger constant*, due to the geometric analogue defined by Cheeger in (Cheeger 1969). Next, we define the Laplacian matrix of a graph and the Rayleigh quotient which are also used throughout this section.

**Definition 3** (Laplacian matrix). *For a graph $\mathcal{G} = (\mathcal{V}, \mathcal{E})$ of $n$ nodes. The Laplacian matrix $\boldsymbol{L}$ is defined as $\boldsymbol{L} = \boldsymbol{D} - \boldsymbol{A}$, where $\boldsymbol{D}$ is the degree matrix and $\boldsymbol{A}$ is the adjacency matrix.*

**Definition 4** (Rayleigh quotient). *For a given symmetric matrix $\boldsymbol{M} \in \mathcal{R}^{n \times n}$ and non-zero vector $\boldsymbol{a} \in \mathcal{R}^n$, the Rayleigh quotient $R_{\boldsymbol{M}}(\boldsymbol{a})$, is defined as: $R_{\boldsymbol{M}}(\boldsymbol{a}) = \frac{\boldsymbol{a}^\top \boldsymbol{M} \boldsymbol{a}}{\boldsymbol{a}^\top \boldsymbol{a}}$.*

We now define a signed Laplacian matrix.

**Definition 5** (Signed Laplacian matrix). *For a graph $\mathcal{G} = (\mathcal{V}, \mathcal{E})$ of $n$ nodes. A signed Laplacian matrix, $\boldsymbol{M}$, is a symmetric matrix that satisfies $\boldsymbol{x}^\top \boldsymbol{M} \boldsymbol{x} = \sum_{(i,j) \in \mathcal{E}} (y_i x_i - y_j x_j)^2$, where $\boldsymbol{y}$ is an eigenvector of $\boldsymbol{M}$ with eigenvalue 0, and $y_i \in \{+1, -1\}$.*

Note that the typical Laplacian matrix, as in Definition 3, fulfills the conditions of Definition 5 with $y_i = +1$ for all $i$. Next, we present an intermediate result for later use.

**Lemma 1.** *Let $\mathcal{G} = (\mathcal{V}, \mathcal{E})$ be an undirected graph of $n$ nodes with Laplacian $\boldsymbol{L}$. Let $\boldsymbol{M} \in \mathcal{R}^{n \times n}$ be a signed Laplacian with eigenvector $\boldsymbol{y}$ as in Definition 5, and let $\boldsymbol{a} \in \mathcal{R}^n$ be a vector such that $\langle \boldsymbol{y}, \boldsymbol{a} \rangle = 0$. Finally, let $\boldsymbol{1} \in \mathcal{R}^n$ be a vector of ones. Then we have that, for a given $\delta \in \mathcal{R}$, $R_{\boldsymbol{L}}(\boldsymbol{a} \circ \boldsymbol{y} + \delta \boldsymbol{1}) \leq R_{\boldsymbol{M}}(\boldsymbol{a})$, where the operator $\circ$ denotes the Hadamard product.*

*Proof.* First, note that $\boldsymbol{L}$ has a 0 eigenvalue with corresponding eigenvector $\boldsymbol{1}$. Also, we have that $\boldsymbol{x}^\top \boldsymbol{L} \boldsymbol{x} = \sum_{(i,j) \in \mathcal{E}} (x_i - x_j)^2$, for any vector $\boldsymbol{x}$. Then, $(\boldsymbol{a} \circ \boldsymbol{y} + \delta \boldsymbol{1})^\top \boldsymbol{L} (\boldsymbol{a} \circ \boldsymbol{y} + \delta \boldsymbol{1}) = \sum_{(i,j) \in \mathcal{E}} ((y_i a_i + \delta) - (y_j a_j + \delta))^2 = (y_i a_i - y_j a_j)^2 = \boldsymbol{a}^\top \boldsymbol{M} \boldsymbol{a}$. Therefore, we have that the numerators of $R_{\boldsymbol{L}}(\boldsymbol{a} \circ \boldsymbol{y} + \delta \boldsymbol{1})$ and $R_{\boldsymbol{M}}(\boldsymbol{a})$ are equal. For the denominators, one can observe that: $(\boldsymbol{a} \circ \boldsymbol{y} + \delta \boldsymbol{1})^\top (\boldsymbol{a} \circ \boldsymbol{y} + \delta \boldsymbol{1}) = (\boldsymbol{a} \circ \boldsymbol{y}) \top (\boldsymbol{a} \circ \boldsymbol{y}) + 2\delta \langle \boldsymbol{1}, \boldsymbol{a} \circ \boldsymbol{y} \rangle + \delta^2 \boldsymbol{1}^\top \boldsymbol{1} = \sum_i a_i y_i a_i y_i + 2\delta \langle \boldsymbol{a}, \boldsymbol{y} \rangle + \delta^2 n = \boldsymbol{a}^\top \boldsymbol{a} + \delta^2 n \geq \boldsymbol{a}^\top \boldsymbol{a}$, which implies that $R_{\boldsymbol{L}}(\boldsymbol{a} \circ \boldsymbol{y} + \delta \boldsymbol{1}) \leq R_{\boldsymbol{M}}(\boldsymbol{a})$. $\qquad\square$

In what follows, we present our first result, which has a connection to Cheeger's inequality (Cheeger 1969).

**Theorem 1.** *Let $\mathcal{G}, \boldsymbol{M}, \boldsymbol{L}, \boldsymbol{y}$ be defined as in Lemma 1, and let $\lambda_1 \leq \lambda_2 \leq \cdots \leq \lambda_n$ be the eigenvalues of $\boldsymbol{M}$. Then, we have that $\frac{\phi_{\mathcal{G}}^2}{4 \Delta_{\max}} \leq \lambda_2$.*

*Proof.* Since $\boldsymbol{y}$ is an eigenvector of $\boldsymbol{M}$ with eigenvalue 0, and $\boldsymbol{M}$ is a symmetric matrix, we can express $\lambda_2$ using the variational characterization of eigenvalues as follows:

$$\lambda_2 = \min_{\boldsymbol{a} \in \mathcal{R}^n,\ \boldsymbol{a}^\top \boldsymbol{y} = 0} R_{\boldsymbol{M}}(\boldsymbol{a}), \tag{4}$$

where we used the fact that $\boldsymbol{y}$ is orthogonal to all the other eigenvectors, by the Spectral Theorem.

Assume that $\boldsymbol{a}$ is the eigenvector associated with $\lambda_2$, i.e., we have that $\boldsymbol{M} \boldsymbol{a} = \lambda_2 \boldsymbol{a}$ and $\boldsymbol{a}^\top \boldsymbol{y} = 0$. Then, by Lemma 1, we have that:

$$R_{\boldsymbol{L}}(\boldsymbol{a} \circ \boldsymbol{y} + \delta \boldsymbol{1}) \leq R_{\boldsymbol{M}}(\boldsymbol{a}) = \lambda_2. \tag{5}$$

Next, we choose $\delta \in \mathcal{R}$ such that $\{a_1 y_1 + \delta, a_2 y_2 + \delta, \ldots, a_n y_n + \delta\}$ has median 0. The reason for the zero median is to later ensure that the subset of vertices $\mathcal{S}$ has less than $n/2$ vertices. Let $\boldsymbol{w} = \boldsymbol{a} \circ \boldsymbol{y} + \delta \boldsymbol{1}$. From equation (5), we have that $R_{\boldsymbol{L}}(\boldsymbol{w}) \leq \lambda_2$.

Let $\boldsymbol{w}^+ = (w_i^+) \top$ such that $w_i^+ = w_i$ if $w_i \geq 0$ and $w_i^+ = 0$ otherwise. Let $\boldsymbol{w}^- = (w_i^-) \top$ such that $w_i^- = w_i$ if $w_i \leq 0$ and $w_i^- = 0$ otherwise. Then, we have that either $R_{\boldsymbol{L}}(\boldsymbol{w}^+) \leq 2 R_{\boldsymbol{L}}(\boldsymbol{w})$ or $R_{\boldsymbol{L}}(\boldsymbol{w}^-) \leq 2 R_{\boldsymbol{L}}(\boldsymbol{w})$. Now suppose that w.l.o.g. $R_{\boldsymbol{L}}(\boldsymbol{w}^+) \leq 2 R_{\boldsymbol{L}}(\boldsymbol{w})$, then, it follows that $R_{\boldsymbol{L}}(\boldsymbol{w}^+) \leq 2\lambda_2$.

Let us scale $\boldsymbol{w}^+$ by some constant $\beta \in \mathcal{R}$ so that: $\{\beta w_1, \beta w_2, \ldots, \beta w_m\} \subseteq [0, 1]$. It is clear that $R_{\boldsymbol{L}}(\boldsymbol{w}^+) = R_{\boldsymbol{L}}(\beta \boldsymbol{w}^+)$, therefore, we will still use $\boldsymbol{w}^+$ to denote the rescaled vector. That is, now the entries of vector $\boldsymbol{w}^+$ are in between 0 and 1.

Next, we will show that there exists a set $\mathcal{S} \subset \mathcal{V}$ with $|\mathcal{S}| \leq n/2$ such that: $\frac{\mathbb{E}[|\mathcal{E}(\mathcal{S}, \mathcal{S}^C)|]}{\mathbb{E}[|\mathcal{S}|]} \leq \sqrt{2 R_{\boldsymbol{L}}(\boldsymbol{w}^+) \Delta_{\max}}$. We construct the set $\mathcal{S}$ as follows. We choose $t \in [0, 1]$ uniformly at random and let $\mathcal{S} = \{i \mid (w_i^+)^2 \geq t\}$. Let $B_{i,j} = 1$ if $i \in \mathcal{S}$ and $j \in \mathcal{S}^C$ or if $j \in \mathcal{S}$ and $i \in \mathcal{S}^C$, and $B_{i,j} = 0$

otherwise. Then, $\mathbb{E}[|\mathcal{E}(\mathcal{S},\mathcal{S}^C)|] = \mathbb{E}[\sum_{(i,j)\in\mathcal{E}} \mathsf{B}_{i,j}] = \sum_{(i,j)\in\mathcal{E}} \mathbb{E}[\mathsf{B}_{i,j}] = \sum_{(i,j)\in\mathcal{E}} P((w_j^+)^2 \leq t \leq (w_i^+)^2)$.

Recall that $(w_i^+)^2 \in [0,1]$, therefore, the probability above is $|(w_i^+)^2 - (w_j^+)^2|$. Thus,

$$\mathbb{E}[|\mathcal{E}(\mathcal{S},\mathcal{S}^C)|] = \sum_{(i,j)\in\mathcal{E}} |w_i^+ - w_j^+| \, |w_i^+ + w_j^+| \leq \sqrt{\sum_{(i,j)\in\mathcal{E}} (w_i^+ - w_j^+)^2} \sqrt{\sum_{(i,j)\in\mathcal{E}} (w_i^+ + w_j^+)^2} \tag{6}$$

$$\leq \sqrt{\sum_{(i,j)\in\mathcal{E}} (w_i^+ - w_j^+)^2} \sqrt{2\sum_{(i,j)\in\mathcal{E}} (w_i^+)^2 + (w_j^+)^2} \leq \sqrt{\sum_{(i,j)\in\mathcal{E}} (w_i^+ - w_j^+)^2} \sqrt{2\Delta_{\max}\sum_i (w_i^+)^2}, \tag{7}$$

where eq.(6) is due to Cauchy-Schwarz inequality and eq.(7) uses the maximum-degree of a node for an upper bound.

Now consider another random variable $\mathsf{b}_i$ such that $\mathsf{b}_i = 1$ if $i \in \mathcal{S}$, and $\mathsf{b}_i = 0$ otherwise. Therefore, we have that $\mathbb{E}[|\mathcal{S}|] = \mathbb{E}[\sum_i \mathsf{b}_i] = \sum_i \mathbb{E}[\mathsf{b}_i] = \sum_i P(t \leq (w_i^+)^2) = \sum_i (w_i^+)^2$. Thus, $\frac{\mathbb{E}[|\mathcal{E}(\mathcal{S},\mathcal{S}^C)|]}{\mathbb{E}[|\mathcal{S}|]} \leq \frac{\sqrt{\sum_{(i,j)\in\mathcal{E}}(w_i^+-w_j^+)^2}\sqrt{2\Delta_{\max}\sum_i(w_i^+)^2}}{\sum_i(w_i^+)^2} = \frac{\sqrt{\sum_{(i,j)\in\mathcal{E}}(w_i^+-w_j^+)^2}\sqrt{2\Delta_{\max}}}{\sqrt{\sum_i(w_i^+)^2}} = \sqrt{2R_L(\boldsymbol{w}^+)\Delta_{\max}} \leq 2\sqrt{\lambda_2\Delta_{\max}}$. The above implies that there exists some $\mathcal{S}$ such that $\frac{|\mathcal{E}(\mathcal{S},\mathcal{S}^C)|}{|\mathcal{S}|} \leq 2\sqrt{\lambda_2\Delta_{\max}}$. Therefore, $\phi_{\mathcal{G}} \leq 2\sqrt{\lambda_2\Delta_{\max}}$ or equivalently $\frac{\phi_{\mathcal{G}}^2}{4\Delta_{\max}} \leq \lambda_2$. □

**Remark 1.** *For a given undirected graph $\mathcal{G}$, its Laplacian matrix $\boldsymbol{L}$ fulfills the conditions of Lemma 1 and Theorem 1. That is, if $\boldsymbol{M} = \boldsymbol{L}$ in Theorem 1 then it becomes the known Cheeger's inequality. Therefore, our result in Theorem 1 apply for more general matrices and is of use for our next result.*

We now provide the SDP relaxation of problem (3). Let $\boldsymbol{Y} = \boldsymbol{y}\boldsymbol{y}^\top$, we have that $\boldsymbol{y}^\top \mathbf{X}\boldsymbol{y} = \mathrm{Tr}(\mathbf{X}\boldsymbol{Y}) = \langle \mathbf{X}, \boldsymbol{Y}\rangle$. Since our prediction is a column vector $\boldsymbol{y}$, we have that $\boldsymbol{y}\boldsymbol{y}^\top$ is rank-1 and symmetric, which implies that $\boldsymbol{Y}$ is a positive semidefinite matrix. Therefore, our relaxation to the combinatorial problem (3) results in the following primal formulation[2]:

$$\max_{\boldsymbol{Y}} \quad \langle \mathbf{X}, \boldsymbol{Y}\rangle \quad \text{subject to} \quad Y_{ii} = 1, \ \boldsymbol{Y} \succeq 0. \tag{8}$$

We will make use of the following matrix concentration inequality for our main proof.

**Lemma 2** (Matrix Bernstein inequality, Theorem 1.4 in (Tropp 2012)). *Consider a finite sequence $\{\mathbf{N}_k\}$ of independent, random, self-adjoint matrices with dimension $n$. Assume that each random matrix satisfies $\mathbb{E}[\mathbf{N}_k] = 0$ and $\lambda_{\max}(\mathbf{N}_k) \leq R$ almost surely. Then, for all $t \geq 0$, $P\left(\lambda_{\max}\left(\sum_k \mathbf{N}_k\right) \geq t\right) \leq n \cdot \exp\left(\frac{-t^2/2}{\sigma^2 + Rt/3}\right)$, where $\sigma^2 = \|\sum_k \mathbb{E}[\mathbf{N}_k^2]\|$.*

The next theorem includes our main result and provides the conditions for exact recovery of labels with high probability.

**Theorem 2.** *Let $\mathcal{G} = (\mathcal{V},\mathcal{E})$ be an undirected connected graph with $n$ nodes, Cheeger constant $\phi_{\mathcal{G}}$, and maximum node degree $\Delta_{\max}$. Then, for the combinatorial problem (3), a solution $\boldsymbol{y} \in \{\boldsymbol{y}^*, -\boldsymbol{y}^*\}$ is achievable in polynomial time by solving the SDP based relaxation (8), with probability at least $1 - \epsilon_1(\phi_{\mathcal{G}}, \Delta_{\max}, p)$, where $p$ is the edge noise from our model, and*

$$\epsilon_1(\phi_{\mathcal{G}}, \Delta_{\max}, p) = 2n \cdot e^{\frac{-3(1-2p)^2\phi_{\mathcal{G}}^4}{1536\Delta_{\max}^3 p(1-p) + 32(1-2p)(1-p)\phi_{\mathcal{G}}^2\Delta_{\max}}}.$$

*Proof.* Without loss of generality assume that $\boldsymbol{y} = \boldsymbol{y}^*$. The first step of our proof corresponds to finding sufficient conditions for when $\boldsymbol{Y} = \boldsymbol{y}\boldsymbol{y}^\top$ is the unique optimal solution to SDP (8), for which we make use of the Karush-Kuhn-Tucker (KKT) optimality conditions (Boyd & Vandenberghe 2004). In the following we write the dual formulation of SDP (8):

$$\min_{\boldsymbol{V}} \quad \mathrm{Tr}(\boldsymbol{V}) \quad \text{subject to} \quad \boldsymbol{V} \succeq \mathbf{X}, \boldsymbol{V} \text{ is diagonal.} \tag{9}$$

Thus, we have that $\boldsymbol{Y} = \boldsymbol{y}\boldsymbol{y}^\top$ is guaranteed to be an optimal solution under the following conditions:

1. $\boldsymbol{y}\boldsymbol{y}^\top$ is a feasible solution to the primal problem (8).

2. There exists a matrix $\boldsymbol{V}$ feasible for the dual formulation (9) such that $\text{Tr}(\mathbf{X}\boldsymbol{y}\boldsymbol{y}^\top) = \text{Tr}(\boldsymbol{V})$.

The first point is trivially verified. For the second point, we assume strong duality in order to find a dual certificate. To achieve that, we make $\mathbf{V}_{i,i} = (\mathbf{X}\boldsymbol{Y})_{i,i}$.[3] If $\mathbf{V} - \mathbf{X} \succeq 0$ then the matrix $\mathbf{V}$ is a feasible solution to the dual formulation. Thus, our first condition is to have $\mathbf{V} - \mathbf{X} \succeq 0$, and we conclude that $\boldsymbol{y}\boldsymbol{y}^\top$ is an optimal solution to SDP (8).

For showing that $\boldsymbol{y}\boldsymbol{y}^\top$ is the unique optimal solution, it suffices to have $\lambda_2(\mathbf{V} - \mathbf{X}) > 0$. Suppose that $\widehat{\boldsymbol{Y}}$ is another optimal solution to SDP (8). Then, from complementary slackness we have that $\langle \mathbf{V} - \mathbf{X}, \widehat{\boldsymbol{Y}} \rangle = 0$, and from primal feasibility $\widehat{\boldsymbol{Y}} \succeq 0$. Moreover, notice that we have $(\mathbf{V} - \mathbf{X})\boldsymbol{y} = 0$, i.e., $\boldsymbol{y}$ is an eigenvector of $\mathbf{V} - \mathbf{X}$ with eigenvalue 0. By assumption, the second smallest eigenvalue of $\mathbf{V} - \mathbf{X}$ is greater than 0, therefore, $\boldsymbol{y}$ spans all of its null space. This fact combined with complementary slackness, primal and dual feasibility, entail that $\widehat{\boldsymbol{Y}}$ is a multiple of $\boldsymbol{y}\boldsymbol{y}^\top$. Thus, we must have that $\widehat{\boldsymbol{Y}} = \boldsymbol{y}\boldsymbol{y}^\top$ because $\widehat{Y}_{i,i} = 1$.

From the points above we arrived to the two following sufficient conditions:

$$\mathbf{V} - \mathbf{X} \succeq 0 \quad \text{and} \quad \lambda_2(\mathbf{V} - \mathbf{X}) > 0. \tag{10}$$

Our next step is to show when condition (10) is fulfilled with high probability. Since we have that $\boldsymbol{y}$ is an eigenvector of $\mathbf{V} - \mathbf{X}$ with eigenvalue zero, showing that $\lambda_2(\mathbf{V} - \mathbf{X}) > 0$ will imply that $\mathbf{V} - \mathbf{X}$ is positive semidefinite. Therefore, we focus on controlling its second smallest eigenvalue. Next, we have that:

$$
\begin{aligned}
\lambda_2(\mathbf{V} - \mathbf{X}) > 0 \quad &\Longleftrightarrow \quad \lambda_2(\mathbf{V} - \mathbf{X} - \mathbb{E}[\mathbf{V} - \mathbf{X}] + \mathbb{E}[\mathbf{V} - \mathbf{X}]) > 0 \\
&\Longleftarrow \quad \lambda_1(\mathbf{V} - \mathbb{E}[\mathbf{V}]) + \lambda_1(\mathbb{E}[\mathbf{X}] - \mathbf{X}) + \lambda_2(\mathbb{E}[\mathbf{V} - \mathbf{X}]) > 0.
\end{aligned}
\tag{11}
$$

We now focus on condition (11) since it implies that $\lambda_2(\mathbf{V} - \mathbf{X}) > 0$. For the first two summands of condition (11) we make use of Lemma 2, while for the third summand we make use of Theorem 1. From $\mathbf{V}_{i,i} = (\mathbf{X}\boldsymbol{Y})_{i,i}$, we have that $\mathbf{V}_{i,i} = y_i \mathbf{X}_{i,:}\boldsymbol{y}$, thus, $\mathbf{V}_{i,i} = \sum_{j=1}^n y_i y_j \mathbf{X}_{i,j} = \sum_{j=1}^n \mathrm{z}_p^{(i,j)} \mathbb{1}\big[(i,j) \in \mathcal{E}\big]$. Then, its expected value is: $\mathbb{E}[\mathbf{V}_{i,i}] = \Delta_i (1 - 2p)$.

**Bounding the third summand of condition** (11). Our goal is to find a non-zero lower bound for the second smallest eigenvalue of $\mathbb{E}[\mathbf{V} - \mathbf{X}]$. Notice that $\mathbb{E}[\mathbf{V} - \mathbf{X}] \succeq 0$ since it is a diagonally dominant matrix, and $\boldsymbol{y}$ is its first eigenvector with eigenvalue 0, i.e., $\lambda_1(\mathbb{E}[\mathbf{V} - \mathbf{X}]) = 0$.

Then, we write $\boldsymbol{M} = \mathbb{E}[\mathbf{V} - \mathbf{X}]$. Now we focus on finding a lower bound for $\lambda_2(\boldsymbol{M})$. We use the fact that for any vector $\boldsymbol{a} \in \mathcal{R}^n$, we have that $\boldsymbol{a}^\top \boldsymbol{M} \boldsymbol{a} = (1 - 2p) \sum_{(i,j) \in \mathcal{E}} (y_i a_i - y_j a_j)^2$.

We also note that $\boldsymbol{M}$ has a 0 eigenvalue with eigenvector $\boldsymbol{y}$. Thus, the matrix $\boldsymbol{M}/(1-2p)$ satisfies the conditions of Theorem 1 and we have that $\lambda_2(\boldsymbol{M}/(1-2p)) \geq \frac{\phi_{\mathcal{G}}^2}{4\Delta_{\max}}$. We conclude that,

$$\lambda_2(\mathbb{E}[\mathbf{V} - \mathbf{X}]) \geq (1 - 2p)\frac{\phi_{\mathcal{G}}^2}{4\Delta_{\max}}. \tag{12}$$

**Bounding the first summand of condition** (11). Let $\mathbf{N}_p^{(i,j)} = \mathrm{z}_p^{(i,j)}(\boldsymbol{e}_i \boldsymbol{e}_i^\top + \boldsymbol{e}_j \boldsymbol{e}_j^\top)$, where $\boldsymbol{e}_i$ is the standard basis, i.e., the vector of all zeros except the $i$-th entry which is 1. We can now write $\mathbf{V} = \sum_{(i,j) \in \mathcal{E}} \mathbf{N}_p^{(i,j)}$. Then, we have a sequence of independent random matrices $\{\mathbb{E}[\mathbf{N}_p^{(i,j)}] - \mathbf{N}_p^{(i,j)}\}$, where we obtain the following: $\lambda_{\max}(\mathbb{E}[\mathbf{N}_p^{(i,j)}] - \mathbf{N}_p^{(i,j)}) \leq 2(1-p)$, and also $\|\sum_{(i,j) \in \mathcal{E}} \mathbb{E}[(\mathbb{E}[\mathbf{N}_p^{(i,j)}] - \mathbf{N}_p^{(i,j)})^2]\| \leq 4\Delta_{\max} p(1-p)$.

Next, we use the fact that $\lambda_{\max}(\boldsymbol{A}) = -\lambda_1(-\boldsymbol{A})$ for any matrix $\boldsymbol{A}$. Then, by applying Lemma 2, we obtain:

$$P\Big(\lambda_1\big(\mathbf{V} - \mathbb{E}[\mathbf{V}]\big) \leq \frac{-(1-2p)\phi_{\mathcal{G}}^2}{8\Delta_{\max}}\Big) \leq n \cdot e^{\frac{-3(1-2p)^2\phi_{\mathcal{G}}^4}{1536\Delta_{\max}^3 p(1-p) + 32(1-2p)(1-p)\phi_{\mathcal{G}}^2\Delta_{\max}}} \tag{13}$$

**Bounding the second summand of condition** (11). Using similar arguments to the concentration above, we now analyze $\lambda_1(\mathbb{E}[\mathbf{X}] - \mathbf{X})$. Let $\mathbf{H}^{(i,j)} = \mathrm{X}_{i,j}(\boldsymbol{e}_i\boldsymbol{e}_j^\top + \boldsymbol{e}_j\boldsymbol{e}_i^\top)$. Then, we have a sequence of independent random matrices $\{\mathbf{H}^{(i,j)} - \mathbb{E}[\mathbf{H}^{(i,j)}]\}$ and we can write $\mathbf{X} = \sum_{(i,j)\in\mathcal{E}} \mathbf{H}^{(i,j)}$. Finally, we have that $\lambda_{\max}(\mathbf{H}^{(i,j)} - \mathbb{E}[\mathbf{H}^{(i,j)}]) \leq 2(1-p)$, and $\mathbb{E}[(\mathbf{H}^{(i,j)} - \mathbb{E}[\mathbf{H}^{(i,j)}])^2] = 4p(1-p)(\boldsymbol{e}_i\boldsymbol{e}_i^\top + \boldsymbol{e}_j\boldsymbol{e}_j^\top)$. Thus, $\|\sum_{(i,j)\in\mathcal{E}} \mathbb{E}[(\mathbf{H}^{(i,j)} - \mathbb{E}[\mathbf{H}^{(i,j)}])^2]\| \leq 4\Delta_{\max}p(1-p)$ and by applying Lemma 2 we obtain:

$$P\left(\lambda_1\big(\mathbb{E}[\mathbf{X}] - \mathbf{X}\big) \leq \frac{-(1-2p)\phi_\mathcal{G}^2}{8\Delta_{\max}}\right) \leq n \cdot e^{\frac{-3(1-2p)^2\phi_\mathcal{G}^4}{1536\Delta_{\max}^3 p(1-p)+32(1-2p)(1-p)\phi_\mathcal{G}^2\Delta_{\max}}} \qquad (14)$$

Note that the thresholds in the concentrations above are motivated by equation (12). Finally, combining equations (12), (13), and (14), we have that:

$$P\big(\lambda_2(\mathbf{V} - \mathbf{X}) > 0\big) \geq 1 - 2ne^{\frac{-3(1-2p)^2\phi_\mathcal{G}^4}{1536\Delta_{\max}^3 p(1-p)+32(1-2p)(1-p)\phi_\mathcal{G}^2\Delta_{\max}}},$$

which concludes our proof. $\qquad\square$

Regarding the statistical part from Theorem 2, it is natural to ask under what conditions we obtain a high probability statement. For example, one can observe that if $\phi_\mathcal{G}^2/\Delta_{\max} \in \Omega(n)$ then there is an exponential decay in the probability of error. Another example would be if $\Delta_{\max} \in \mathcal{O}(\sqrt{n})$ and $\phi_\mathcal{G}^2/\Delta_{\max} \in \Omega(\sqrt{n})$ then we also obtain high probability argument. Thus, we are interested in finding what classes of graphs fulfill these or other structural properties so that we obtain a high probability bound in Theorem 2. Regarding the computational complexity of exact recovery, from Theorem 2, we are solving a SDP, and any SDP can be solved in polynomial time using methods such as the interior point method.

## 3.2 Second Stage

After the first stage, we obtain two feasible solutions for problem (3), that is, $\boldsymbol{y} \in \{\boldsymbol{y}^*, -\boldsymbol{y}^*\}$. To decide which solution is correct we will use the node observations $\boldsymbol{c}$. Specifically, we will output the vector $\boldsymbol{y}$ that maximizes the score $\boldsymbol{c}^\top\boldsymbol{y}$. The next theorem formally states that, with high probability, $\boldsymbol{y} = \boldsymbol{y}^*$ maximizes the score $\boldsymbol{c}^\top\boldsymbol{y}$ for a sufficiently large $n$.

**Theorem 3.** *Let $\boldsymbol{y} \in \{\boldsymbol{y}^*, -\boldsymbol{y}^*\}$. Then, with probability at least $1 - \epsilon_2(n,q)$, we have that: $\boldsymbol{c}^\top\boldsymbol{y}^* = \max_{\boldsymbol{y}\in\{\boldsymbol{y}^*, -\boldsymbol{y}^*\}} \boldsymbol{c}^\top\boldsymbol{y}$, where $\epsilon_2(n,q) = e^{-\frac{n}{2}(1-2q)^2}$ and $q$ is the node noise.*

The remaining proofs of our manuscript can be found in Appendix A.

**Remark 2.** *From Theorems 2 and 3, we obtain that exact recovery (i.e., $\boldsymbol{y} = \boldsymbol{y}^*$) is achievable with probability at least $1 - \epsilon_1(\phi_\mathcal{G}, \Delta_{\max}, p) - \epsilon_2(n,q)$. Finally, from Theorem 3, it is clear that since the parameter $q \in (0, 0.5)$, for a sufficiently large $n$ we have an exponential decay of the probability of error $\epsilon_2$. Thus, we focus on the conditions of the first stage and provide examples in the next section.*

## 4 Examples of Graphs for Exact Recovery

In this section, we provide examples of classes of graphs that yield high probability in Theorem 2.

Perhaps the most important example we provide in this section is related to the smoothed analysis on connected graphs (Krivelevich et al. 2015). Consider any fixed graph $\mathcal{G} = (\mathcal{V}, \mathcal{E})$ and let $\widetilde{\mathcal{E}}$ be a random set of edges over the same set of vertices $\mathcal{V}$, where each edge $e \in \widetilde{\mathcal{E}}$ is independently drawn according to the Erdős-Rényi model with probability $\varepsilon/n$ and where $\varepsilon$ is a small (fixed) positive constant. We denote this as $\widetilde{\mathcal{E}} \sim \text{ER}(n, \varepsilon/n)$, then let $\widetilde{\mathcal{G}} = (\mathcal{V}, \mathcal{E} \cup \widetilde{\mathcal{E}})$ denote the random graph with the edge set $\widetilde{\mathcal{E}}$ added.

The model above can be considered a generalization of the classical Erdős-Rényi random graph, where one starts from an empty graph (i.e., $\mathcal{G} = (\mathcal{V}, \emptyset)$) and adds edges between all possible pairs of vertices independently with a given probability. The focus on "small" $\varepsilon$ means that we are interested in the effect of a rather gentle random perturbation. In particular, it is known that graphs with bad expansion are not suitable for exact inference (see for instance, (Abbe et al. 2014)), but certain classes

such as grids or planar graphs can yield good approximation under some regimes despite being bad expanders as shown by Globerson et al. (2015). Here we consider the graph $\mathcal{G}$ to be a bad expander and show that with a small perturbation, exact inference is achievable.

The following result was presented by (Krivelevich et al. 2015) in an equivalent fashion.[4]

**Lemma 3** (Theorem 2 in (Krivelevich et al. 2015)). *Let $\mathcal{G} = (\mathcal{V}, \mathcal{E})$ be a connected graph, choose $\widetilde{\mathcal{E}} \sim \mathrm{ER}(n, \varepsilon/n)$, and let $\widetilde{\mathcal{G}} = (\mathcal{V}, \mathcal{E} \cup \widetilde{\mathcal{E}})$. Then, for every $\varepsilon \in [1, n]$, we have that $\phi_{\widetilde{\mathcal{G}}} \geq \frac{\varepsilon}{256 + 256 \log n}$, with probability at least $1 - n^{-2.2 - \frac{\log \varepsilon}{2}}$.*

The above lemma allows us to lower bound the Cheeger constant of the random graph $\widetilde{\mathcal{G}}$ with high probability, and is of use for our first example.

**Corollary 1.** *Let $\mathcal{G} = (\mathcal{V}, \mathcal{E})$ be any connected graph, choose $\widetilde{\mathcal{E}} \sim \mathrm{ER}(n, \log^8 n/n)$, let $\widetilde{\mathcal{G}} = (\mathcal{V}, \mathcal{E} \cup \widetilde{\mathcal{E}})$ and let $\Delta_{\max}^{\widetilde{\mathcal{G}}}$ be the maximum node degree of $\widetilde{\mathcal{G}}$. Then, we have that $\phi_{\widetilde{\mathcal{G}}}^2/\Delta_{\max}^{\widetilde{\mathcal{G}}} \in \Omega(\log^5 n)$ and $\Delta_{\max}^{\widetilde{\mathcal{G}}} \in \mathcal{O}(\log^9 n)$ with high probability. Therefore, exact recovery in polynomial time is achievable with high probability.*

We emphasize the nice property of random graphs $\widetilde{\mathcal{G}}$ shown in Corollary 1, that is, by adding a small perturbation (edges from the Erdős-Rényi model with small probability) we are able to obtain exact inference despite of $\mathcal{G}$ having bad properties such as being a bad expander. Our next two examples include complete graphs and $d$-regular expanders. The following corollary shows that, with high probability, exact recovery of labels for complete graphs is possible in polynomial time.

**Corollary 2** (Complete graphs). *Let $\mathcal{G} = \mathcal{K}_n$, where $\mathcal{K}_n$ denotes a complete graph of $n$ nodes. Then, we have that $\phi_{\mathcal{G}}^2/\Delta_{\max} \in \Omega(n)$. Therefore, exact recovery in polynomial time is achievable with high probability.*

Another important class of graphs that admits exact recovery is the family of $d$-regular expanders (Hoory et al. 2006), which is defined below.

**Definition 6** ($d$-regular expander). *A $d$-regular graph with $n$ nodes is an expander with constant $c > 0$ if, for every set $\mathcal{S} \subset \mathcal{V}$ with $|\mathcal{S}| \leq n/2$, $|\mathcal{E}(\mathcal{S}, \mathcal{S}^C)| \geq c \cdot d \cdot |\mathcal{S}|$.*

**Corollary 3** (Expanders graphs). *Let $\mathcal{G}$ be a $d$-regular expander with constant $c$. Then, we have that $\phi_{\mathcal{G}}^2/\Delta_{\max} \in \Omega(d)$. If $d \in \Omega(\log n)$ then exact recovery in polynomial time is achievable with high probability.*

## 5 Concluding Remarks

We considered a model where we receive a single noisy observation for each edge and each node of a graph. Our approach consisted of two stages, similar in spirit to (Globerson et al. 2015). The first stage consisted of solving solely the quadratic term of the optimization problem and was based in a SDP relaxation in order to find the structural properties of a graph that guarantee exact recovery with high probability. Given two solutions from the first stage, the second stage consisted in using solely the node observations and simply outputting the vector with higher score. We showed that for any graph $\mathcal{G}$, the term $\phi_{\mathcal{G}}^2/\Delta_{\max}$ is related to achieve exact recovery in polynomial time. Examples include complete graphs and $d$-regular expanders, that are guaranteed to recover the correct labeling with high probability. While perhaps the most interesting example is related to smoothed analysis on connected graphs, where even for a graph with bad properties such as bad expansion can still be exactly recovered by adding small perturbations (edges coming from an Erdős-Rényi model with small probability).

## Acknowledgments

This material is based upon work supported by the National Science Foundation under Grant No. 1716609-IIS.

## Footnotes

[1] In the literature, the cost functions $c_v$ and $c_{u,v}$ are also known as unary and pairwise potentials respectively.

[2]Here we dropped the constant ½ since it does not change the decision problem.

[3]Note that we now write $\mathbf{V}$ in upright shape (i.e., $\mathbf{V}$) since it contains randomness from $\mathbf{X}$.

[4] Specifically, we set $\alpha = 1/2, \delta = \varepsilon/256$, $K = 128/\varepsilon$, $C = 1$, $s = K \log n$, which results with all the conditions being fulfilled in the proof of Theorem 2 in (Krivelevich et al. 2015).

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
