[Supplementary Material · NeurIPS19-exact-supp.pdf]



# SUPPLEMENTARY MATERIAL
## Exact inference in structured prediction

## A  Detailed Proofs

In this section, we state the proofs of Theorem 3 and Corollaries 1, 2, 3 from our manuscript.

### A.1  Proof of Theorem 3

*Proof.* We are interested in upper bounding the probability of predicting the wrong vector $\boldsymbol{y}$, that is,

$$
\begin{aligned}
P(\boldsymbol{c}^\top \boldsymbol{y}^* \leq -\boldsymbol{c}^\top \boldsymbol{y}^*) &= P(\boldsymbol{c}^\top \boldsymbol{y}^* \leq 0) \\
&= P\Big( \sum_{u \in \mathcal{V}} z_q^{(u)} \leq 0 \Big) \\
&\leq e^{-\frac{n}{2}(1-2q)^2},
\end{aligned}
$$

where for the last equation we applied Hoeffding's inequality.  □

### A.2  Proof of Corollary 1

*Proof.* Fix $\varepsilon = \log^8 n$. Let $\epsilon_r(n,\varepsilon) = n^{-2.2 - \frac{\log \varepsilon}{2}}$, then from Lemma 3 we get $\phi_{\widetilde{\mathcal{G}}} \in \Omega(\log^7 n)$ with probability at least $1 - \epsilon_r(n,\varepsilon)$. Let $\Delta_{\max}$ be the maximum node degree of graph $\mathcal{G}$, then it is clear that $\Delta_{\max}^{\widetilde{\mathcal{G}}}$ is a random variable with expected value $\mathbb{E}[\Delta_{\max}^{\widetilde{\mathcal{G}}}] \leq \Delta_{\max} + \log^8 n$. By applying Markov's inequality we obtain $P(\Delta_{\max}^{\widetilde{\mathcal{G}}} \geq t) \leq \mathbb{E}[\Delta_{\max}^{\widetilde{\mathcal{G}}}]/t \leq (\Delta_{\max} + \log^8 n)/t$ for $t > 0$. Set $t = \log^9 n$, then let $\epsilon_\Delta(\Delta_{\max}, n) = (\Delta_{\max} + \log^8 n)/\log^9 n$, we have that $\Delta_{\max}^{\widetilde{\mathcal{G}}} \leq \log^9 n$ with probability at least $1 - \epsilon_\Delta(\Delta_{\max}, n)$.

By using the union bound and noting that $\epsilon_r \to 0$ and $\epsilon_\Delta \to 0$ as $n \to \infty$, we have that $\phi_{\widetilde{\mathcal{G}}}^2 / \Delta_{\max}^{\widetilde{\mathcal{G}}} \in \Omega(\log^5 n)$ and $\Delta_{\max}^{\widetilde{\mathcal{G}}} \in \mathcal{O}(\log^9 n)$ with high probability. Finally, this leads to $\epsilon_1 \to 0$ as $n \to \infty$, thus, exact inference is achievable in polynomial time.  □

### A.3  Proof of Corollary 2

*Proof.* For any set $\mathcal{S} \subset \mathcal{V}$ with $|\mathcal{S}| \leq n/2$, we have that:

$$
\phi_{\mathcal{S}} = \frac{|\mathcal{E}(\mathcal{S}, \mathcal{S}^C)|}{|\mathcal{S}|} = \frac{|\mathcal{S}| \cdot |\mathcal{S}^C|}{|\mathcal{S}|} = |\mathcal{S}^C| \quad \implies \quad \phi_{\mathcal{G}} = \lceil \frac{n}{2} \rceil.
$$

Since $\mathcal{G}$ is a complete graph, we have that $\Delta_{\max} = n - 1$, which yields $\phi_{\mathcal{G}}^2 / \Delta_{\max} \in \Omega(n)$. Thus, from Theorem 2, we have that $\epsilon_1(\phi_{\mathcal{G}}, \Delta_{\max}, p) \to 0$ as $n \to \infty$.  □

### A.4  Proof of Corollary 3

*Proof.* From Definition 6, we have that $\phi_{\mathcal{G}} \geq c \cdot d$. Since the graph is regular, we have that $\Delta_{\max} = d$. Therefore, $\phi_{\mathcal{G}}^2 / \Delta_{\max} \in \Omega(d)$. Finally, if $d \in \Omega(\log n)$, then $\epsilon_1(\phi_{\mathcal{G}}, \Delta_{\max}, p)$ decays in at least $n^{-c_1}$ for some constant $c_1 > 0$. That is, $\epsilon_1(\phi_{\mathcal{G}}, \Delta_{\max}, p) \to 0$ as $n \to \infty$.  □