[Reviews · NeurIPS 2019]

Reviewer 1



Detailed remarks Concerning theorem 1, the bound you prove is related to theorem 4.3 of (Mohar, B. (1989). Isoperimetric numbers of graphs. Journal of combinatorial theory) where a tighter upper bound is proved for laplacian matrix. It would be great to discuss whether your bound is tight in the general case of signed Laplacian matrices knowing that this one is (up to what I read) is tight for the Laplacian one. In particular it would be great to recall what you refer to as the 'Cheeger inequality' (which I believe concerns the first smallest eigenvalue) and the known result for the 2nd eigenvalue of the Laplacian in the paper previously cited Concerning the derivation of theorem 2, the result of theorem 1 appears naturally as the ingredient to bound the 3rd term which is the main originality of the proof. As a consequence, I think it would improve the readability of the paper to announce the use of Th1 earlier in the paper since I understood the logic of your sections only at my second read of the paper. Finally it would have been great to provide a numerical illustration of your result in theorem 2 by simulating graphs and showing whether the bound of theorem 2 indicating when the graph is correctly recovered by the SDP. This could have improved the paper even if it is not mandatory for the sake of the presentation. Minor remarks The notation change from V to \textit{V} (footnote (3)) is a bit weird since it is very hard to distinguish.

Reviewer 2



Overview: - This paper studies the conditions for exact recovery of ground-truth labels in structured prediction under some data generation assumptions. In particular, the analysis generalizes the one in Globerson et al. (2015) from grid graphs to general connected graphs, providing high-probability guarantees for exact label recovery which depend on structural properties of the graph. - On the one hand, extending the results of Globerson et al. (2015) to general graphs seems like a valid contribution. On the other hand, the assumed generative process (lines 89-101, proposed in Globerson et al., 2015) is somewhat toyish which might make the results less interesting. Therefore, I am inclined towards acceptance but not strongly. - I went over most of the proofs in detail and did not find obvious errors (see more minor comments below though). Comments: - I feel like the presentation can be greatly improved by including an overview of the main result at the beginning of Section 3. In particular, you can state the main result, which is actually given in Remark 2 (!), and then provide some high-level intuition on the path to prove it. This may mean that some derivations will have to be deferred to the appendix, which I think is fine. - Lack of empirical evaluation is another obvious shortcoming of this paper. - Perhaps you can come up with a title that is more specific to this paper (e.g., “Exact Recovery of Labels in Structured Prediction with General Graphs”)? Additional related work: - Very recent paper (published after NeurIPS submission): Approximate Inference in Structured Instances with Noisy Categorical Observations, Heidari et al., to appear in UAI 2019. Generalizes Globerson et al. (2015) to multiclass variables. I am wondering if you can use their result to generalize your approach from binary to multiclass variables. - Result on tractable models (line 41): Tightness of LP relaxations for almost balanced models, Weller et al., AISTATS (2016). - Result on approximate inference (line 47): Train and test tightness of LP relaxations in structured prediction, Meshi et al., ICML (2016). - Very recent result on label recovery (published after NeurIPS submission): Accuracy-Memory Tradeoffs and Phase Transitions in Belief Propagation, Jain et al., COLT (2019). The generative process seems very different and the focus is Belief Propagation, so this may be only remotely related. Other minor comments: - Line 142: Let M be a signed Laplacian **with eigenvector y** as in Definition 5. Otherwise it’s not clear what y is in the statement of Lemma 1. - Line 163: missing superscript before \top - Line 167: did you mean “\beta w_i^+” instead of just “\beta w_i”? - Line 175: I am probably missing something, but why is the probability equal to |(w_i^+)^2 - (w_j^+)^2|? Don’t you also have to require that (w_i^+)^2 \geq (w_j^+)^2? - Line 197: consider adding n to the statement of Thm 2 as it appears in the expression for epsilon. This is actually done in Thm 3, so makes sense here as well. Also, maybe mention in Thm 3 that q is the node noise (as done in Thm 2 for p)? - Line 274: \mathcal{R} is overloaded, used earlier for Reals. - Lines 282-286: what does “bad expansion” mean?

Reviewer 3



This paper is a, quite technical, theoretical paper that addresses a variant of a tough and old combinatoric problem : the (exact) recovery of a ground state from a noisy planted spin-glass model recast here as a structured prediction inference problem. It comes as a follow-up from two papers : "How hard is inference for structured prediction" by (Globerson et al. 2015), and "Inference in Sparse Graphs with Pairwise Measurements and Side Information" by (Foster et al. 2018). The key contribution of this paper is to answer an open problem stated in (Globerson et al. 2015) by providing a characterisation of the graphs where the exact recovery is tractable. The authors define the hardness of the exact recovery in term of a probability bound : the probability of recovering the exact ground state in polynomial time is higher when the graph is a good expander (i.e. when its Cheeger constant is high) and lower when its nodes have high degrees. They provide examples of graph families like the "d-regular expanders" that fulfil this requirement. As a by-product they provide an extension of Cheeger inequality to signed Laplacians. I found the paper reasonably clear but not as well written as (Globerson et al. 2015) for instance. It lacks a proper justification, a few explaining figures and concrete examples to guide the reader before she/he dives into the (painful) technical aspects of the proofs. The state of the art is really minimalistic: see "Related Work" section in (Globerson et al. 2015). It's not mandatory but it would make sense to move at last one of the proofs in the appendix (keeping only a sketch in the main paper) and use the freed space to fix these issues. I only deeply checked the proof of Theorem 1, but the rest of the maths (mainly the proof of Theorem 2) seems solid. Minor remarks/questions: - The edges noise model as defined on line 105 is a noisy observation of edges from correct spins, would it be easier for instance if we considered a generative model where all the edges would be derived from the same noisy observation of the ground node spins ? How would it impact the error bounds ? - It's only a factor 2 but please make it clear what kind of node degree you use in the Laplacian (oriented or non-oriented) POST REBUTTAL: If the degree of the D matrix is the non-oriented degree i.e. the number of edges connected, then the Laplacian should be L=2D-A otherwise x^T L x is not as stated in Definition 5.

[Author Response · NeurIPS 2019]

We thank the reviewers for their comments and feedback. For all reviewers,
we plan to add some experiments to show empirical correctness of our
results. Fig. 1 is a plot for complete graphs, where for each number of
vertices, 30 runs were computed in order to estimate the probability of
success. The ground truth $y^*$ was generated uniformly at random and we
used $p = 0.2$ and $q = 0.2$. Standard CVX code was used for SDP.

**R1:** • We appreciate the reference of B. Mohar (1989), regarding tightness,
our bound is tight up to a factor of $1/2$. This can be achieved by considering a
signed Laplacian matrix $\mathbf{M}$ with first eigenvector $y = \mathbf{1}$ and first eigenvalue
0, then we obtain a bound for the typical Laplacian matrix, whose known
tight lower bound for $\lambda_2$ is $\frac{\phi_{\mathcal{G}}^2}{2\Delta_{\max}}$. See for instance section 3 in F. R. K. Chung (1996), "Laplacians of graphs and
Cheeger inequalities" (where $\Delta_{\max}$ does not appear in their bound due to their different definition of Laplacian matrix,
which is already weighted by $1/\Delta_{\max}$). Comparing to Theorem 4.2 in B. Mohar (1989), "Isoperimetric numbers of
graphs", their bound is tighter but has a dependency on $\lambda_2$, i.e., the lower bound is $\frac{\phi_{\mathcal{G}}^2 + \lambda_2^2}{2\Delta_{\max}}$, thus, while interesting, the
proof requires a lower bound independent of $\lambda_2$. We propose to add these brief comments in Section 5.

• For better presentation we plan to replace the start of Section 3 with these lines: "Our approach consists of two
stages, similar in spirit to (Globerson et al., 2015). We first use only the quadratic term from (2), which will give us
two possible solutions with high probability, as stated in Theorem 2. Then as a second stage, the linear term is used to
decide with high probability the best between these two solutions, as stated in Theorem 3. Combining both theorems,
we find the sufficient graph properties to achieve exact recovery."

**R2:** • We think it is unfair to say that the model is toyish and makes results less interesting. While the model is simple,
it deserves attention as it provides further theoretical understanding to structure prediction. Additionally, Globerson et
al. (2015) and Foster et al. (2018) use the same model and have been published in venues such as ICML and AISTATS,
conferences that are similar in spirit to NeurIPS.

• Reviewer 1 also suggested to improve the start of Section 3. Please see bullet 2 in answers to R1.

• We appreciate the new reference (Heidari et al., 2019). The generalization from binary to multiclass, under our
approach, remains as an open question for us. We believe it deserves careful thinking and might even be a conference
paper on its own. The references of Weller et al. (2016), and Meshi et al. (2016), while not close to our work, will
be added to have a more complete overview of results in approximate inference.

• Line 167: Yes, we will fix this typo, it should say $\beta w_i^+$.

• Line 175: In this case we want to know the probability of $i \in \mathcal{S}$ and $j \in \mathcal{S}^C$, or **viceversa** (note this), for $(i, j) \in \mathcal{E}$.
Then, by construction of $\mathcal{S}$, for this to happen, one $w^+$ should be greater than $t$ and the other less than $t$. Since $t$ is a
uniform random variable in $[0, 1]$, the probability is just the difference of the squared values, i.e., $(w_i^+)^2 - (w_j^+)^2$ (see
line 173-174). Then, the absolute value makes sure that if $(w_j^+)^2 > (w_i^+)^2$ then the difference is positive. This would
count the case in which $i \in \mathcal{S}^C$ and $j \in \mathcal{S}$.

• Expansion in graphs is related to algebraic connectivity. The Cheeger constant is one notion of expansion, where a low
Cheeger constant means that the graph has poor algebraic connectivity. One example of graphs that can be considered
bad expanders are grid graphs (as stated in lines 284-285). We will add this notion of expansion before using it in order
to make our arguments more clear.

**R3:** • The model proposed by R3 would indeed be simpler resulting in no need for SDP or other sophisticated
methods/algorithms. This is because one would only have the noisy information from the nodes, and if edges are
derived from these noisy nodes then the edge values are still consistent with the noisy nodes. Thus, edges provide no
further information in R3's suggested model. The hard part of our setting is the inconsistency between noisy edges and
noisy nodes, creating the need for solving a complex combinatorial problem.

• We mention in Section 2, lines 89-101, as well as in the Theorem statements that the graphs are undirected. Therefore,
the node degrees are non-oriented.

• It is hard to talk about $\Delta_{\max}$ or $\phi_{\mathcal{G}}$ in isolation and might not be the way to look at the bounds. Thus we focus on
family of graphs and provide examples in Section 4. We also mention cases where the guarantee does not work, for
example, for grid graphs (lines 284-285). This is the reason why we discuss smoothed complexity.

• We appreciate the reference suggestion "Statistical physics of inference: Thresholds and algorithms" by Zdeborova
and Krzakala 2018, it contains a comprehensive review of inference problems and we plan to add it in our paper.

[Meta-Review · NeurIPS 2019]

The paper gives a theoretical analysis of Markov random fields. The authors answer the question of when exact inference can be done exactly in a polynomial time. This is a generalization of a result of in Globerson et al. (2015) from grid graphs to general connected graphs, which is on my opinion, a non-trivial generalization. The paper is self contained and readable for the Machine Learning community, although quite technical. Indeed, I consider that it is a theoretical paper that has all the quality for a NeurIPS acceptance. Note however that the technical aspect of the paper leads me to simply recommend a poster, and not a talk.